



# Stability requirements of observation systems to detect long-term stratospheric ozone trends based upon Monte Carlo simulations

Mark Weber[1]

[1]Institut für Umweltphysik, Universität Bremen FB1, Bremen, Germany
**Correspondence:** Mark Weber (weber@uni-bremen.de)

**Abstract.** For new observing systems, particularly satellites, specifications on the stability required for climate variables are provided in order to be useful for certain applications, for instance, deriving long-term trends. The stability is usually stated in units of percent per decade (%/dec) and is often associated with or termed instrument drift. A stability requirement of 3%/dec or better has been recently stated for tropospheric and stratospheric ozone. However, the way this number is derived is not
clear. In this study we use Monte Carlo simulations to investigate how a stability requirement translates into uncertainties of long-term trends depending on the lifetime of individual observing systems, which are merged into timeseries, and the period of available observations. Depending on the need to observe a certain trend over a given period, e.g. typically +1%/dec for total ozone and +2%/dec for stratospheric ozone over thirty years, stability for observation systems can be properly specified and justified in order to achieve statistical significance in the observed long-term trend. Assuming a typical mean lifetime of
seven years for an individual observing system and a stability of 3%/dec results in a 2%/dec trend uncertainty over a period of 30 years, which is barely sufficient for stratospheric ozone but too high for total ozone. Having two or three observing systems simultaneously time reduces the uncertainty by 30% and 42%, respectively. The method presented here is applicable to any variable of interest for which long-term changes are to be detected.

## 1   Introduction

Ninety percent of ozone in the overhead (total) column resides in the stratosphere and protects the earth system from harmful UV radiation (Zerefos et al., 2023). The phase-out of ozone-depleting substances according to the Montreal Protocol and its Amendments, lead to recovery of stratospheric ozone in some region of the atmosphere after the late 1990s (Hassler et al., 2022). Most significant trends are observed in the upper stratosphere by up to about +2%/dec (Godin-Beekmann et al., 2022). In the extratropics total column ozone has increased by a rate of up to about +1%/dec (Coldewey-Egbers et al., 2022; Weber
et al., 2022). The reported statistical trend uncertainty at $2\sigma$ level for ozone profiles are typically of the same order than the trend itself, about 1-2%/dec, similarly for total column ozone at about 0.5-1%/dec (LOTUS, 2019; Godin-Beekmann et al., 2022; Weber et al., 2018, 2022).

Uncertainties in ozone trends reported recently (e.g. Hassler et al., 2022; Godin-Beekmann et al., 2022; Weber et al., 2022) are determined from the statistics of trend regression alone. Bourassa et al. (2014) reported added uncertainties due to drifts of
the OSIRIS satellite to the statistical uncertainty, here 3%/dec, which reduced the atmospheric region (altitude and latitude) of



the atmosphere showing significant positive trends. Direct comparison between various merged ozone profiles show that the drifts in the difference between zonal mean single-instrument datasets are on the order of 3%/dec but not always statistically significant (Rahpoe et al., 2015; Hubert et al., 2016). Similarly, the spread in the recent total ozone trends from the available merged datasets are on the order of ±0.5%/dec, which are indicative of drifts between the merged datasets. The apparent drifts are not only due to changes in the instrument performance but can also dependent on the way how the individual datasets are merged into the long-term dataset (Frith et al., 2017; Weber et al., 2022). Since multiple time series are available, the statistical trend uncertainty can be further reduced by averaging trends or calculating trends from the the median or mean of datasets (e.g. Steinbrecht et al., 2017; LOTUS, 2019; Godin-Beekmann et al., 2022; Weber et al., 2022).

Ozone is expected to recover from the decrease in ozone-depleting substances but long-term changes also depend on the evolution of greenhouse gases (GHG) as well as on the feedback mechanism between ozone and climate (Hassler et al., 2022) The Vienna Convention for the Protection of the Ozone Layer (VIENNA, 1985), which set among others the framework for the Montreal Protocol signed in 1987, also stated the need for continuing observation of ozone, related species, and climate gases. For long-term observations of ozone and other trace species, specific requirements on accuracy and stability for observing systems were defined and updated over the years (e.g. GCOS-01, 2001; IGACO-04, 2004; GCOS-11, 2011; CCI, 2021; CMUG, 2022; GCOS-22, 2022). For trend detection a stability requirement of 3%/dec were stated initially for stratospheric ozone and 1%/dec for total ozone (GCOS-01, 2001). Main application area defined were trends and operational meteorology for stratospheric ozone (GCOS-01, 2001). The 3%/dec value is also currently defined as the threshold value for stratospheric and total column ozone, meaning that observing system will be only useful if this threshold is not exceeded (CCI, 2021; GCOS-22, 2022; CMUG, 2022). In addition to the threshold value, a breakthrough (2%/dec) and target value (1%/dec) is provided (GCOS-22, 2022).

It is not clear how the specification are derived, but very likely they are leaned from the observed ozone statistical trend uncertainties and the expectation that the requirements should be close but lower than these uncertainties. The question we rise here in the paper is how such a requirement can be justified from the need to achieve a certain trend uncertainty over a given period of time. From the ozone recovery perspective the time range since ODS peaked in the stratosphere is close to thirty years. A similar but different question was addressed by Weatherhead et al. (1998). Based upon the noise of data and serial correlation, they determined on statistical grounds how many years we need to observe a significant trend, which is the case when twice the trend uncertainty is lower than the trend magnitude ($2\sigma$ significance).

We use Monte Carlo simulation to investigate how a stability requirement translates into uncertainties of long-term trends depending on the lifetime of individual observing systems, merged into timeseries, and the period of available observations. In Section 2 the scheme of the Monte Carlo simulation is described. Results are presented in Section 3, and discussed in Section 4. We close with concluding remarks in Section 5.





## 2   Monte Carlo simulations

The scheme of Monte Carlo simulations of time series and derived trend uncertainties due to a given stability requirement is shown in Fig. 1 (e.g. Guimarães Couto et al., 2013). We assume that a simulated timeseries without any drift and bias after subtraction of the start value is zero. Each individual observing system with a given lifetime (segment) of which we compose a long-term merged dataset (timeseries) has varying drifts in %/dec, which follow a Gaussian distribution with a given $1\sigma$ stability requirement, e.g. 1.5%/dec as shown in panel a. The variability seen in the sample of one million long-term timeseries (panel b) are therefore solely due to the instrument drifts following the stability distribution. As an example four single timeseries over 30 years are shown in red. Each segment is seven years long corresponding to a constant lifetime. For simplicity, an overlap period is not considered here (Weatherhead et al., 2017).

For each of the timeseries a trend was determined and its distribution of the sample is shown in panel c. Here the trends are calculated for a thirty year period. A stability requirement of 1.5%/dec and a life time of 7 years for each observing system results in a trend uncertainty of 0.77%/dec after 30 years (close to half the value of the stability requirement). With a required limit of stability of 3%/dec ($2\sigma$) for ozone, trends lower than about 1.5%/dec are not detectable after 30 years (simply doubling the numbers given in the previous sentence).

Some additional modifications in the simulation were introduced. In a merged dataset the individual observing systems have varying lifetimes. An expected lifetime of seven years is typical for current satellite systems, e.g. TROPOMI (Veefkind et al., 2012). In a different setup we therefore vary the lifetimes of individual segments following a Poisson distribution with a mean of seven years (Fig. 2). When merging datasets, usually segments of a single observation system with some extended records are used. In our case we set a lower limit of five years to be included in the timeseries.

Additional uncertainties come from biases between individuals segments. Biases can be corrected when sufficient overlap exists between observing systems as discussed in detail in Weatherhead et al. (2017). In our simulations we do not consider any overlap periods, but we allow for a Gaussian distributed bias from one segment to the other. Figure 3 shows the samples of timeseries assuming 0, 1, and 2% bias ($1\sigma$), respectively. In both simulations varying lifetimes with a mean of 7 years and Poisson distributed (Fig. 2) were assumed.

With a stability requirement of 1.5%/dec the trend uncertainty after 30 years increases slightly from 0.77 to 0.83%/dec using varying lifetimes instead of a constant lifetime of seven years (Figs. 2b and 3a). A larger change in uncertainties is seen by adding a 1% bias between segments, resulting into an increase to 1.1%/dec. This means that for ozone observed with a 3%/dec stability in observing systems and a typical bias of 1% between them, the trend uncertainty is 2.2%/dec over 30 years.

## 3   Results

The dependence of the trend uncertainty as a function of observation years assuming a stability of 1.5%/dec is shown in Fig. 4. The different colours in the plot represents lifetimes for the individual observing systems varying from 5 to 20 years. For a given stability requirement and period of the timeseries, the trend uncertainties increases with lifetime. This is on first sight somewhat unexpected as we normally desire to have observing or satellite missions to operate as long as possible. A continuous





**Figure 1.** Monte-Carlo simulation of 30-year trend uncertainties assuming a stability requirement of 3%/dec ($2\sigma$) for individual observing systems with a lifetime of seven years. Panel a: Gaussian instrumental drift distribution assuming a stability requirement of 3%/dec ($2\sigma$). Panel b: Ten thousand timeseries (blue lines) from the one million sample of simulated timeseries, composed of seven-year individual segments (single instrument observations), are shown. Red lines show four examples of timeseries from the sample. Panel c: Distribution of linear trends derived from all timeseries.

90   drift over a longer time period will cause larger deviations of the timeseries from the truth and increases trend uncertainties. It is also evident that the reduction in trend uncertainties with time slows considerably after two decades.

So far we considered the case of one single mission contributing to the time series. If we have several parallel missions with identical geographical coverage for long-term monitoring the trend uncertainties can be strongly reduced. To investigate this effect, mean timeseries were constructed by averaging $n$ randomly generated timeseries, with $n$ being interpreted as the

95   number of parallel missions. The result is shown in Fig. 5. Trend uncertainties are reduced by a factor of $\frac{1}{\sqrt{n}}$ from the single mission case. With two and three missions in parallel over the entire period, the trend uncertainty is reduced by nearly 30% and



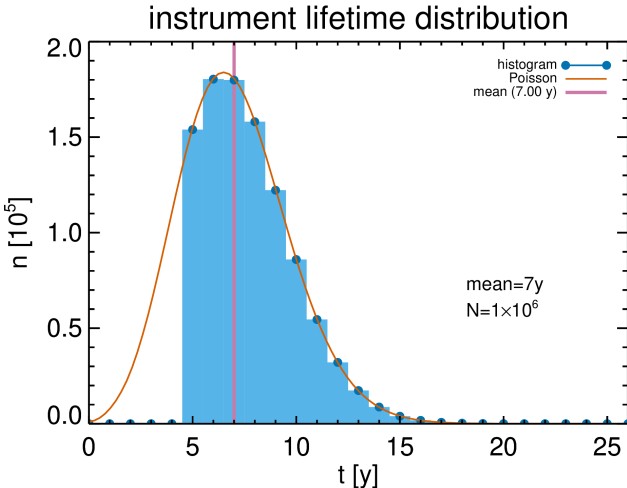

**Figure 2.** Distribution of lifetimes for single observing systems based upon a Poisson distribution with a mean of seven years. Segments with lifetimes of less than 5 years are not allowed in the timeseries and the distribution is set to zero below five years.

42%, respectively. Redundancy in observing systems is therefore more effective in reducing trend uncertainties than improving the stability of an observing system. This is an important consideration in a long-term monitoring program (Harris et al., 2015).

Figure 6 displays the impact of biases between segments on trend uncertainty as a function of the timeseries length. Here we assume again the 1.5%/dec stability and results are shown for variable lifetimes with a mean of 7 (solid) and 12 years (dashed line). If there are no biases (black curves in all panels) then the trend uncertainty decrease faster with time for lower lifetimes as already demonstrated in Fig. 4. If the mean bias gets larger, this reverses and trend uncertainty decreases faster for timeseries with segments having longer lifetimes. With a bias between consecutive observation systems, the trend uncertainty initially increases with time and starts decreasing after a couple years. A too large bias causes the trend uncertainty to be always larger than the stability requirement (red and blue curves in panel a). One of the requirements for long-term monitoring is to keep biases between consecutive observing systems as low as possible. Biases can be considerably reduced when the overlap period of two observation systems or satellite missions is sufficiently long as discussed in Weatherhead et al. (2017). Redundancy in observing systems again helps to reduce effect from the biases.

## 4 Discussion

We present here a simple approach of Monte Carlo simulations to specify stability requirements dependent on the need to observe a certain trend in merged datasets after a given period of time. Stratospheric ozone is expected to recover since the end of the 1990s (somewhat less than thirty years ago) when stratospheric halogens released from ozone-depleting substances reached maximum. A rule of thumb is that after thirty years the trend uncertainty is about 2%/dec assuming a typical mean lifetime of 7 years and a stability of 3%/dec of the individual missions or observing systems of which the timeseries is composed



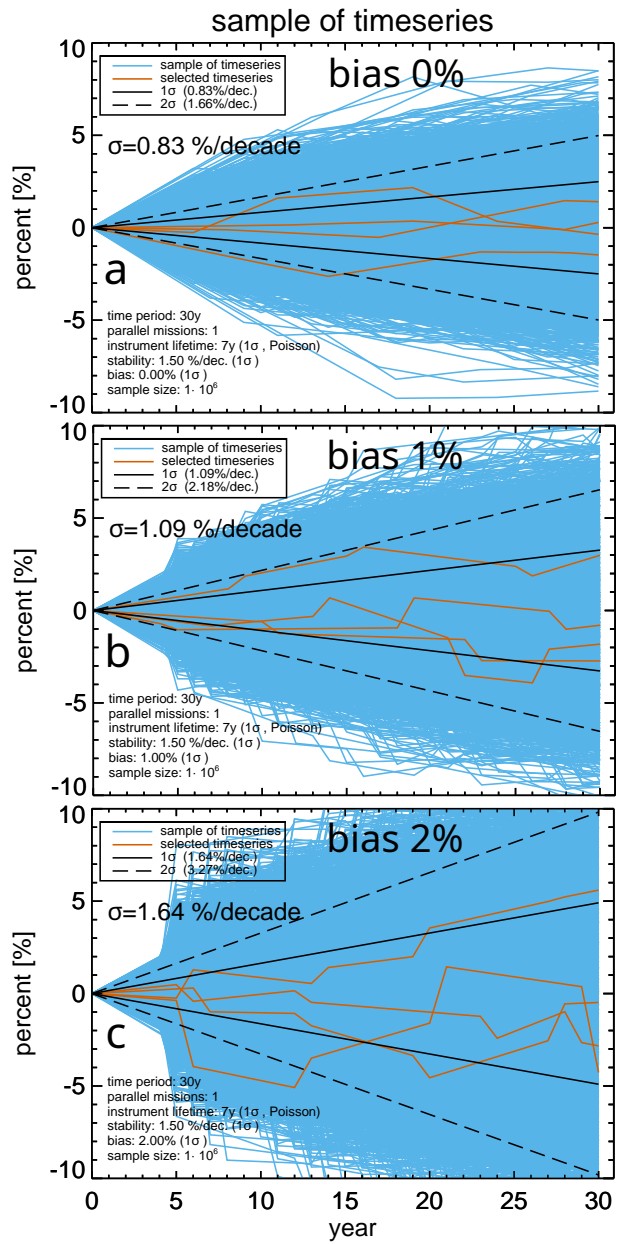

**Figure 3.** Samples of 30-year timeseries composed of segments with varying lifetimes following a Poisson distribution with a mean and standard deviation of 7 years. Ten thousand timeseries (blue lines) from the one million sample of simulated timeseries, composed of seven-year individual segments (single instrument observations), are shown in each panel. Red lines show four examples of timeseries from the sample. Panel a: no bias between segments; panel b and c: biases drawn from a Gaussian distribution with $\sigma = 1\%$ and 2%, respectively.

115     of. If the bias between consecutive observing systems is negligible due to a good bias correction in case of a sufficient overlap between consecutive missions, the uncertainty improves to about 1.5%/dec.





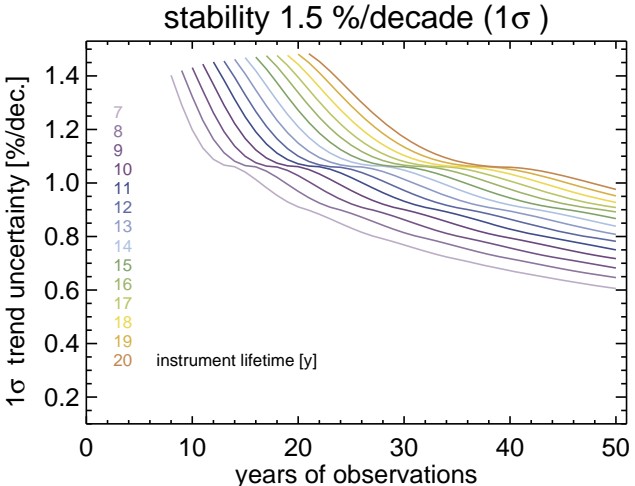

**Figure 4.** Trend uncertainties as a function of observation period assuming a stability requirement of 1.5%/dec for individual segments with a constant lifetime. Different colours show results for lifetimes varying from 5 to 20 years.

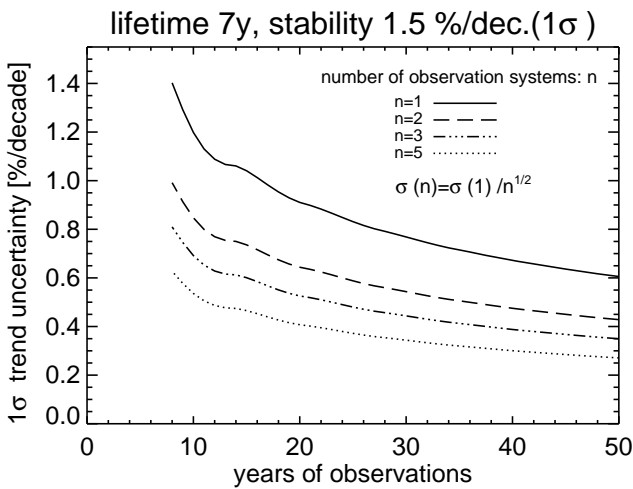

**Figure 5.** Trend uncertainties as a function of observation period and number of parallel observing systems ($n =1$, 2, 3, and 5) assuming a constant lifetime of 7 years and a stability of 1.5%/dec ($1\sigma$) for each system.

Our simple approach of course neglects the larger likelihood that instrument drifts with time are not necessarily linear nor drifts are normal distributed. Largest changes in satellite performance is usually in the beginning or near the end of the mission. When the ambient environment of a satellite instrument changes from the clean room at the ground to space conditions shortly after launch, calibration settings changes and if these changes are not adequately accounted for in the trace gas retrieval, causes data drifts. In the first few years instruments in space outgas that change the optical performance of the instrument more rapidly than in later years. In general drifts can be caused by any changes in the optical performance (e.g. Bourassa et al., 2014). For





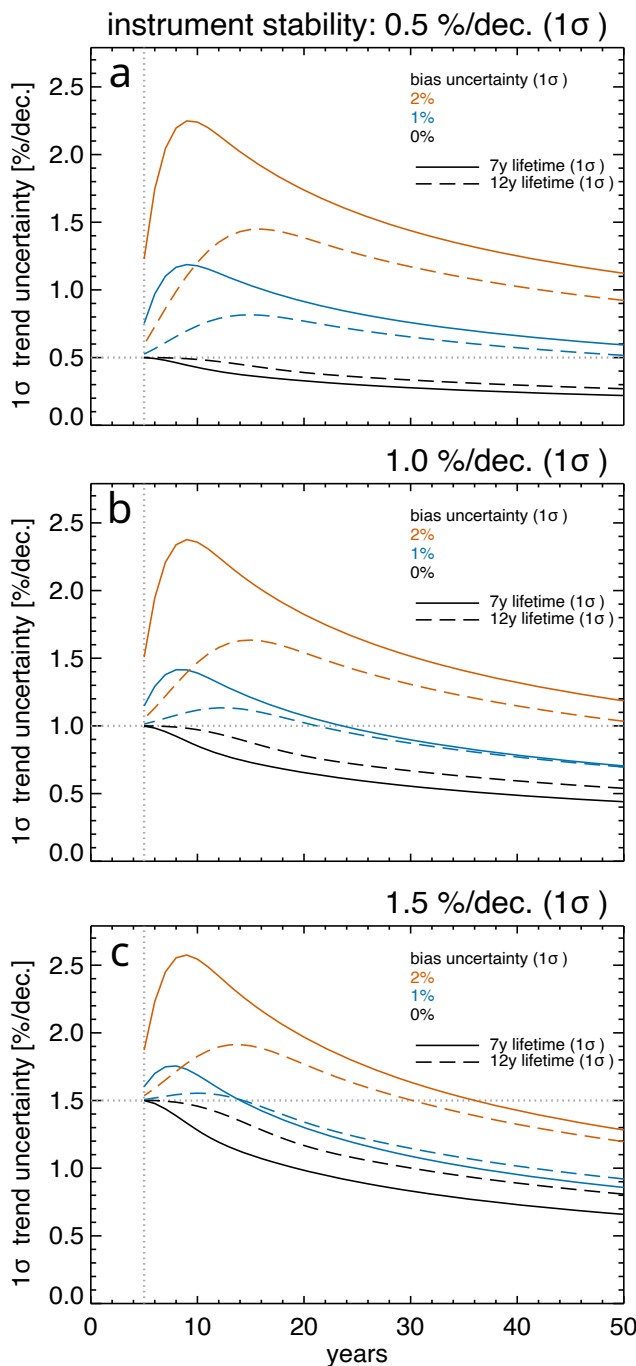

**Figure 6.** Trend uncertainty in %/dec as a function of observation years for various stability requirements (panels a to c), different mean lifetimes of single observing systems (solid and dashed lines), and biases between consecutive observation systems (colors).



limb based satellite data pointing errors to the tangent height can increase with time, particularly in the later part of the mission period (Bourassa et al., 2018). Near the end of the mission, instruments are aged and suffer from loss in thermal and/or power stability which may enhance again the instrumental drift in the observational data.

The multiple merged satellite ozone data records are usually not entirely independent from each other, as the same satellite missions are used in different merged timeseries (e.g. Steinbrecht et al., 2017; LOTUS, 2019; Godin-Beekmann et al., 2022; Weber et al., 2022). In such a case the trend uncertainty reduces more slowly than with $1/\sqrt{n}$, which is only valid for $n$ truly independent timeseries.

## 5 Conclusions

Using a simple Monte Carlo timeseries simulation scheme, we were able to estimate the required stability of an observing instrument with a given lifetime that will be needed to reach a certain trend uncertainty in a long-term timeseries consisting of many of those systems or series of satellites. This requirement does not only depend on the length of the timeseries and lifetime, but also on the biases between consecutive observing systems. With a typical mean lifetime of about 7 years for an observing system, in particular satellites, a 3%/dec stability (GCOS-22, 2022) adds about 1.5%/dec uncertainty to the statistical long-term trend uncertainty after 30 years. For total ozone the GCOS requirement is insufficient and must be higher and on the order of 1%/dec.

The time range of the ozone recovery phase is approaching 30 years in the next few years. Assuming a mean residual bias of about 1% between consecutive measurement systems (after overlap corrections), the resulting long-term trend uncertainty increases to 2%/dec over this period. For the next decades the long-term trend uncertainty will only decrease rather slowly, so that redundancy in observing systems is very effective in reducing the impact from instrument drifts. Uncertainties due to instrument drifts can be cut nearly in half with three parallel observing systems.



*Code and data availability.* Codes in IDL and simulated data are available upon request to the author.

*Author contributions.* The single author did all the work.

*Competing interests.* The author is associate editor of this journal.

*Acknowledgements.* This work was carried out as part of the OREGANO (Ozone Recovery from Merged Observational Data and Model Analysis) project funded by the European Space Agency under the Contract 4000137112/22/I-AG. The author is also grateful for financial support by the State of Bremen via the University of Bremen.



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
