# Peer review of "Stability requirements of satellites to detect long-term stratospheric ozone trends based upon Monte Carlo simulations"

_EGUsphere, 2023_

## Author Comment (AC1)

**Reply to the reviewer's comments**

Weber, M.: Stability requirements of observation systems to detect long-term stratospheric ozone trends based upon Monte Carlo simulations, EGUsphere [preprint], https://doi.org/10.5194/egusphere-2023-3070, 2024.

Reviewer comments in blue
Replies in black
Line numbers refer to the EGUsphere preprint

**Reviewer #1**

Summary:

The authors perform a simple study to determine the impact of how potential drifts in datasets and biases between datasets can affect derived long-term trends through the use of Monte Carlo simulations. While the scope is fairly narrow, providing a short paper, it is a useful complement to studies that have attempted to assess the impact of the intrinsic quality of datasets on trend studies. These results are useful in the broader context of helping inform high-level requirements on future observing systems. I do not have any major concerns with the paper (mostly some clarifications and grammar changes), but I would like to recommend some additional questions or comparisons for the author that I believe will improve the final messaging. However, I would not consider them required prior to publication and will leave it to the author's discretion on whether to include them in the final manuscript.

Thanks for the positive evaluation.

Comments:

Pg02, Ln028: "Similarly, the spread in the recent total ozone trends from the available merged datasets are on the order of ±0.5%/dec, which are indicative of drifts between the merged datasets. The apparent drifts are not only due to ..." I don't believe it is a given that the differences in trend results come from drifts. Drifts are certainly a possibility, but we have yet to prove it. Perhaps change "indicative of drifts" to "suggest the possibility of drifts" and "apparent drifts" to "potential drifts".

Done. See track-change manuscript.

Pg03, Ln066: "For each of the timeseries a trend was determined and its distribution of the sample ..." Just for the sake of clarity, perhaps instead of saying "a trend was determined" you could say "a simpler linear least squares trend was fit"? At least I am assuming that is how you are calculating the trends used for this study since it is not explicitly stated in the paper. Similarly, I would modify "and its distribution of the sample is  shown ..." with "and its corresponding uncertainty, represented by the

standard deviation of the distribution, is shown …” This would explicitly state what the trend uncertainties you refer to are.

 Done. See track-change manuscript.

Pg03, Ln068: “A stability requirement of 1.5%/dec and a life time of 7 years for each observing system results in a trend uncertainty of 0.77%/dec after 30 years (close to half the value of the stability requirement).” In general, it can be easy to mix up the fact that this study often lists 1σ values but ozone trend uncertainties are often reported as 2σ so it might be good to be more explicit at various points in the paper. Given the context of the sentence that follows, this might be a good one to add “(1σ)” after both “%/dec” units. The last sentence of Section 2 is another good place to make this distinction, perhaps even showing the 2σ trend uncertainty value to better illustrate the point. Lastly, it would also be good to clarify this in the conclusion and abstract.

This is clarified at various places in the manuscript. The 1σ results can be converted to n·σ values by multiplying with n. In Section 1 (Introduction, l45), we add the following to the manuscript: “No specification in these documents is given if the stability requirement is stated as one (1σ) or two standard deviations (2σ). With regard to the observed long-term ozone trends, these values make only sense if they are defined as 2σ (two standard deviations).”

Pg03, Ln072: I am not sure if the word “typical” is best when referring to satellite lifetimes. Naturally many instruments/spacecraft last longer than their stated prime mission lifetime of ~3 years, but there are also those that do not and no mission ever claims beforehand to ensure their mission will last that long. Perhaps “typical” is too strong a phrase and “not uncommon” might be more appropriate.

 I do not think this makes much difference using “typical” or “not uncommon”. Leave as is.

Pg03, Ln075: “In our case we set a lower limit of five years to be included in the timeseries.” On one hand I understand why you did this. However, there is an increasing movement toward the idea of repeatedly launching smaller spacecraft that may not have the long lifetime of a larger one but to do so more frequently. It would be interesting to allow this study to go to shorter lifetimes (such as perhaps 3 years).

Compiling a merged dataset using short single-instrument time series is challenging. As discussed in the paper, quite often, changes in calibration settings (not accounted for) right after launch may cause larger drifts in the early years. Merging data requires sufficiently long overlap periods. The paper does not discuss overlap issues, but we refer here to Weatherhead et al. (2017). There has been no change in the manuscript.

Pg03, Ln087: “For a given stability requirement and period of the timeseries, the trend uncertainties increases with lifetime. … A continuous drift over a longer time period will cause larger deviations of the timeseries from the truth and increases trend uncertainties.” I think this is simply because of the statistical distribution of drifts. Obviously if an instrument can be shown to be more stable, it operating for longer is better. However, in the absence of that guarantee, then more shorter missions is seemingly better. It might be worth noting this distinction/caveat.

This is a good point. A data record can be considered stable if a comparison with a reference dataset, which is considered to be truth, shows negligible drifts. Discussing which reference dataset can be

considered truth is beyond the scope here. No metrological standard exists for atmospheric ozone (not at room temperature).

Pg04, Ln097: "Redundancy in observing systems is therefore more effective in reducing trend uncertainties than improving the stability of an observing system." It may be worth reiterating this point in your conclusions. Although it may also be worth considering the comparison of two observing systems where both lack stability (e.g., 5-10%/dec [2σ]) versus having one observing system but with a better stability (e.g., 2-3%/dec [2σ]).

The focus here is on atmospheric ozone, where trends are well below 5%/dec, so we do not gain much when adding this information.

Pg05, Ln101: "If there are no biases (black curves in all panels) then the trend uncertainty decrease faster with time for lower lifetimes ..." It is also worth noting that, in the absence of biases, the trend uncertainty immediately begins to decrease as soon as more data is collected (unlike the beginning years of data with biases).

Added "... the trend uncertainties of bias-free timeseries immediately decrease with additional years of data".

Pg05, Ln113: "A rule of thumb ..." This would imply this is already some sort of common knowledge. I think it might be better to say something along the lines of, and I'm rephrasing your whole sentence here, "While there are many possible combinations of observing systems, we show that a reasonable assumption of repeated single instruments with a mean lifetime of 7 years and a stability requirement of 3%/dec (2σ) would yield a trend uncertainty of about 2%/dec (2σ) after thirty years of observations." Also, am I correct about those sigma values on your sentence?

We replaced the sentence with your suggestion.

In your conclusions, I think it is useful (as you have done) to reiterate some of your results, but I also think it would be useful to put it in the context of some of the current trend results and what sort of requirements we should be looking for in future measurement systems. If we are looking at trend in the stratosphere of 1-2%/dec, what sort of stability do we need to reliably extract those trends? It is also worth differentiating the upper stratosphere from the lower stratosphere as well as the difference with total column. Lastly, I think, from a messaging perspective, it is worth reiterating that allof these uncertainties stem purely from modeling the impacts of drifts and biases and that the intrinsic data quality of the different data sets (i.e., precision of the measurements) will only add additional uncertainty on top of this.

The abstract and conclusion has been revised.

Editorial Comments:

Nearly all are done; see some specific replies below.

Pg01, Ln016: " ... ozone depleting substances ..." Needs ODS acronym added here for later use

Pg01, Ln017: Change "region" to "regions"

Pg01, Ln018: "Most significant trends are observed ..."

    Do you mean "Most statistically significant recovery trends"?

Pg01, Ln019: Insert a comma after "extratropics"

Pg01, Ln020: Insert "the" between "at" and "2σ"

Pg02, Ln030: "... but can also dependent on the way how the individual datasets are merged ..."

    "... but can also depend on how the individual datasets are merged ..."

Pg02, Ln034: Insert a comma after "substances"

Pg02, Ln040: Insert a comma after "detection"

Pg02, Ln040: Change "... were stated ..." to "... was stated ..."

Pg02, Ln041: Change "area" to "areas"

Pg02, Ln043: Insert "an" before "observing system" and "will be only" should be "will only be"

Pg02, Ln046: Change "... the specification are ..." to "... how these specifications are ..."

Pg02, Ln046: "... they are leaned from the observed ..."

    Do you mean "learned from" or "informed by"?

Use "derived from".

Pg02, Ln047: Change "rise" to "raise"

Pg02, Ln049: Insert a comma after "perspective"

Pg02, Ln053: Change "simulation" to "simulations"

Pg02, Ln054: "... merged into timeseries ..."

    Given the usage of this in the greater context of the sentence, I believe this should say "how [data/observations] are merged into a timeseries"

Change to "which are merged into ..."

Pg02, Ln055: Insert a comma after "Section 2"

Pg03, Ln059: "We assume that a simulated timeseries without any drift and bias after subtraction of the start value is zero."

> Move "is zero" from the end to "... bias is zero after ..."

Pg03, Ln060: "Each individual observing system with a given lifetime (segment) of which we

compose a long-term merged dataset (timeseries) has varying drifts ..."

> Insert a comma after "(segment)" and after "(timeseries)"

Pg03, Ln066: Insert a comma after "timeseries"

Pg03, Ln071: Insert a comma before "individual"

Leave as is.

Pg03, Ln075: Insert a comma after "we"

Leave as is.

Pg03, Ln083: Replace "into" with "in" and insert a comma after "that"

Pg03, Ln086: Insert a comma before "assuming" and after "1.5%/dec"

Pg03, Ln087: Replace "represents" with "represent"

Pg04, Ln093: Insert a comma after "monitoring"

Pg05, Ln101: Insert a comma after "panels)" and replace "decrease" with "decreases"

Pg06, Ln115: Insert a comma after negligible and the word "the" before the word "case"

no comma after negligible.

Pg07, Ln117: Remove "of course" and insert "are" after "nor"

Pg07, Ln118: Replace "normal" with "normally" and "changes [...] is" with "changes [...] are"

Pg07, Ln120: Insert "it" before "causes"

Pg07, Ln121: Insert a comma after "years" and change "outgas that change" to "outgas, which changes"

Pg09, Ln123: Insert a comma after "data"

Pg09, Ln140: Insert a comma after "decades"

Citation:
https://doi.org/10.5194/egusphere-2023-3070-RC1

**Reviewer #2:**

This paper presents an interesting analysis based on Monte Carlo simulations of the impact of given stability requirement in ozone observing systems on effective trend detection and assessment of ozone recovery. The main parameters used in the simulations are the lifetime of single satellite missions, stability of the missions and their eventual bias. The author concludes that with the current stability requirements given by GCOS-22, ozone recovery is barely detectable 30 years after the peak of ozone depleting substance in the stratosphere. The manuscript is generally well written and well presented. However, there are some issues and recommendations that need to be considered before publication in Atmospheric Measurement Techniques.

Thanks for your positive evaluation.

The study and more specifically its conclusion address mainly the GCOS-22 3%/dec stability requirement. However, 2%/dec breakthrough and 1%/dec target requirements are also mentioned for total ozone. More discussion is needed on the impact of such requirements on total ozone trend uncertainties. The study is mainly based on satellite missions and their drift over their limited lifetime. However, the global ozone observing system includes also ground-based measurements with much longer lifetime, which provides an additional constraint to the evaluation of satellite instrument stability. How can such constraint from ground-based instruments be taken into account in the study, considering also the findings on the reduction uncertainties from parallel observing systems? The study would benefit from a comparison of the parameters considered in the study (drift, bias, lifetime) to actual satellite measurement time series, e.g. of for total ozone. This would enable an assessment of the total ozone measurements system and provide a more concrete assessment of trend detection capabilities as they currently stand.

The GCOS-22 stability requirements for threshold, breakthrough, and target are identical for total ozone columns and profiles. At line 100, we changed the text to "...show in panels a to c, results for various stability requirements of 0.5, 1.0, and 1.5%/dec (1σ), which corresponds to the GCOS-22 threshold, breakthrough, and target requirements assumed to be given as 2σ values". In the caption of Fig. 6, we add the following text: "Panels a to c show trend uncertainties for the GCOS-22 stability requirements of 1%/dec (2σ, target), 2%/dec (breakthrough), and 3%/dec (threshold), respectively. In the conclusion section, we clearly recommend lowering the threshold stability requirement from 3%/dec to 1%/dec (2σ)."

This study mainly focuses on satellite measurements. Ground-based instruments like Brewer and Dobson have much longer lifetimes. Changes in operation procedures and calibration settings may cause drifts and biases for ground-based instruments, and the concept of lifetimes, as used in the simulations, is not useful here. Therefore, we do not use the term "observation system" but rather refer to satellites in this work. The title of the paper has also been modified.

At l. 73, we added: "Several satellite instruments measured for a decade or more (e.g., SAGE II: 21 years, OSIRIS: more than 20 years, OMI: 19 years, GOME-2A: 12 years, SCIAMACHY, MIPAS, GOMOS:

10 years)." In the conclusion section, we specify the impact of drift uncertainty on the overall uncertainty (statistical plus drift). We also recommend here to reduce the threshold requirements from 3%/dec (2σ), as stated in GCOS-22, to 1%/dec for detecting long-term ozone trends.

 P3 l59-60: The sentence starting with "We assume" should be reformulated since a time series cannot be zero.

In the simulations, we neglect the intrinsic (physical) trends in the timeseries, so that the trends determined from the linear regression are solely due to the drifts in the individual single-instrument data and the bias between them. Therefore, the timeseries without biases and drifts is zero (after subtracting an arbitrary starting value.

The simulation set-up leading to Fig. 4 is not completely clear about the considered bias. Is it equal to 0%? What is the explanation of the author about the bump in the curves around 1.1 % trend uncertainty?

The bias between single-instrument data records is zero here. The concept of biases is introduced later in the manuscript. The flattening of the curves (bumps) is related to the indicated lifetimes. For a 7-year lifetime, the flattening occurs after 15 years (2x7 + 1 year) when the third 7-year segment just starts and acts like a dog tail. Smaller bumps are seen when the fourth 7-year segment starts (3x7 + 1 year). Similar patterns are seen for other lifetimes.

Is a drift of 1.5%/dec detectable from current observing system? Could it be corrected? A refined statistical set up could eventually be envisaged based on e.g. Bayesian statistics reducing the probability of strongly drifting times series over the longest observing periods.

Variations in trends derived from different merged time series are indicative of relative drifts between datasets. This is already stated in l.28-30. We also mention here that the drifts may not be only instrument-related but also depend on the merging approach used.

P5 l114-115: the sentence starting with "A rule of thumb…" should be replaced by a table specifying the trend uncertainty from various parameters, e.g. bias stability and lifetime, summarizing results of Fig. 6.

I believe no table is needed here, as Fig. 6 provides the information needed. The numbers provided here are estimates from simplified simulations.

Fig 6. In order to better explain the figure, another version of Fig1 or Fig3 could be shown, including the specified bias between segments.

I do not get the point here. Fig. 6 is an extension of Fig. 3, which shows timeseries for a 30-year period and 1.5%/dec (2σ) stability requirement only, showing results for different mean lifetimes (7 and 12 years), observation periods (up to 50 years), and biases (0-2%). Figure 3 shows selected timeseries (in red) with randomly selected biases between segments.

P7 l120: replace "outgas that change" by "outgas, which changes".  Done.

Citation: https://doi.org/10.5194/egusphere-2023-3070-RC2